# Tuning phenylalanine fluorination to assess aromatic contributions to protein function and stability in cells

Grace D. Galles [1,2,4], Daniel T. Infield[1,4], Colin J. Clark[1], Marcus L. Hemshorn[2], Shivani Manikandan [1], Frederico Fazan[1], Ali Rasouli [3], Emad Tajkhorshid [3], Jason D. Galpin [1], Richard B. Cooley[2], Ryan A. Mehl[2] & Christopher A. Ahern [1] ✉

The aromatic side-chains of phenylalanine, tyrosine, and tryptophan interact with their environments via both hydrophobic and electrostatic interactions. Determining the extent to which these contribute to protein function and stability is not possible with conventional mutagenesis. Serial fluorination of a given aromatic is a validated method in vitro and in silico to specifically alter electrostatic characteristics, but this approach is restricted to a select few experimental systems. Here, we report a group of pyrrolysine-based aminoacyl-tRNA synthetase/tRNA pairs (tRNA/RS pairs) that enable the site-specific encoding of a varied spectrum of fluorinated phenylalanine amino acids in *E. coli* and mammalian (HEK 293T) cells. By allowing the cross-kingdom expression of proteins bearing these unnatural amino acids at biochemical scale, these tools may potentially enable the study of biological mechanisms which utilize aromatic interactions in structural and cellular contexts.

While primarily appreciated for its hydrophobicity, the benzyl side chain of phenylalanine also engages in various types of energetically favorable aromatic interactions (Fig. 1a)[1–11]. Energetically significant aromatic interactions involving phenylalanine are believed to be widespread in nature[12,13]. Based on emerging structural data, they have been increasingly proposed to play important mechanistic roles in ligand recognition[14–19] and protein–protein interactions, both in normal function[20,21] and as a result of clinical mutation[22]. They may also play important roles in membrane anchoring, via attraction of the benzyl side chain to choline lipid headgroups[23], and have recently been proposed to mediate conduction in some ion channels[24]. Testing the specific functional role for aromaticity of the benzyl side chain in an interaction is not possible using conventional mutagenesis since mutation to a non-aromatic natural amino acid also causes concomitant changes in amino acid shape, convoluting data interpretation. Fluorination of the aromatic ring, on the other hand, allows for

redistribution of the electrostatic potential of Phe with only minor steric changes (Fig. 1b); thus allowing the specific testing of the functional relevance of aromaticity. Increasing the number of fluorine substitutions results in a linear decrease in the electrostatic potential for π interactions[25,26].

The site-specific installation of fluorinated phenylalanine amino acids can be accomplished via peptide synthesis and by the direct injection of misacylated nonsense suppressor tRNA[27], the latter of which has been used to functionally validate and quantify aromatic interactions in some membrane proteins, including ion channels[28–33]. These methods, while chemically flexible, cannot be easily scaled for biochemical and structural approaches. Additionally, some natural amino acids have been replaced with fluorinated, non-canonical structural analogs via permissive endogenous synthetases, but these methods are not site-specific, resulting in multiple sites within the protein sequence being modified heterogeneously[34]. On the other

[1]Department of Molecular Physiology and Biophysics, University of Iowa College of Medicine, Iowa City, IA, USA. [2]The GCE4All Research Center, Department of Biochemistry & Biophysics, Oregon State University, Corvallis, OR, USA. [3]Theoretical and Computational Biophysics Group, NIH Center for Macromolecular Modeling and Bioinformatics, Beckman Institute for Advanced Science and Technology, University of Illinois at Urbana-Champaign, Urbana, IL, USA. [4]These authors contributed equally: Grace D. Galles, Daniel T. Infield. ✉e-mail: christopher-ahern@uiowa.edu

hand, nonsense suppression via coevolved tRNA/aminoacyl tRNA-synthetase (tRNA/RS) pairs allows for biochemical-scale production of proteins bearing site-specifically installed non-canonical amino acids (ncAA) in diverse cell systems and even organisms[27,35]. Previously, one such tRNA/RS pair that encodes select fluorinated phenylalanine analogs into proteins was selected by rational design followed by a screening of a single-residue library of 20 variants. This system was shown to be functional in *Escherichia coli* if grown in minimal media without exogenous phenylalanine[36] but the lack of a sufficiently efficient tRNA/RS pair stymies field progress. A method to accurately encode fluorinated Phe ncAAs in complex media and in eukaryotic cells would prove highly valuable for many clinically relevant soluble and membrane protein targets, including the myriad genes that must be expressed and/or studied in mammalian cells[12].

In this work, we describe the identification and validation of a group of synthetases, which we have named Phe$_x$, from the pyrrolysine (Pyl) system[37–39], which are competent for site-specific

encoding of fluorinated di-, tri-, tetra, and penta-fluoro phenylalanine analogs within proteins in both bacterial and mammalian expression systems. We hypothesized that by screening a greater number of tRNA-synthetase (RS) active site variants (-10⁷), using a strategy that enabled enrichment of variants able to selectively incorporate fluorinated phenylalanine derivatives but not phenylalanine, we may identify synthetases with the desired utility and versatility. Intact protein mass spectrometry defines the synthetase and ncAA combinations that enable high-fidelity encoding at the amber suppression site without spurious substitution at natural codons. Finally, we demonstrate biochemical scale encoding in two large (>150 kDa) human membrane proteins of high clinical and therapeutic interest.

## Results

### Encoding fluorinated phenylalanine analogs in *E. coli*
Historical attempts by our groups to screen for robust pyrrolysine (Pyl) synthetases able to incorporate fluorinated phenylalanine analogs were not successful, possibly owing to the close steric resemblance of fluoro-benzene variants. In the present work, we decided to screen a library of Pyl synthetases with para-methyl, tetra-fluoro-phenylalanine (Fig. 2a), in the hope that the additional structural bulk of the methyl at the para position would permit the identification of synthetase's selective against natural amino acids but selective for fluorinated Phe derivatives, perhaps including those lacking the para-methyl group. Two independent rounds (Fig. 2b) of positive and negative screening within a Pyrrolysine-based RS library (see Methods) in *E. coli*[40] resulted in the identification of several synthetases that enabled amber suppression within superfolder GFP (sfGFP_N150TAG) in response to para- methyl, tetra-fluoro Phe (Fig. 2c). Unique synthetases wherein suppression of sfGFP (from GFP signal) was greater than 10 fold above background. As a preliminary assessment of the quality of the evolved RS pairs, experiments to

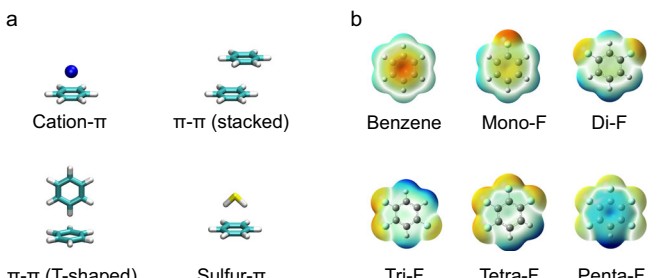

**Fig. 1 | Depictions of geometry and electrostatics. a** Illustrations of types of π interactions in which phenylalanine can participate. **b** Electrostatic views of benzene (as a structural surrogate for phenylalanine) and fluorinated analogs. Coloring is to the convention of Red = negative, Blue = positive.

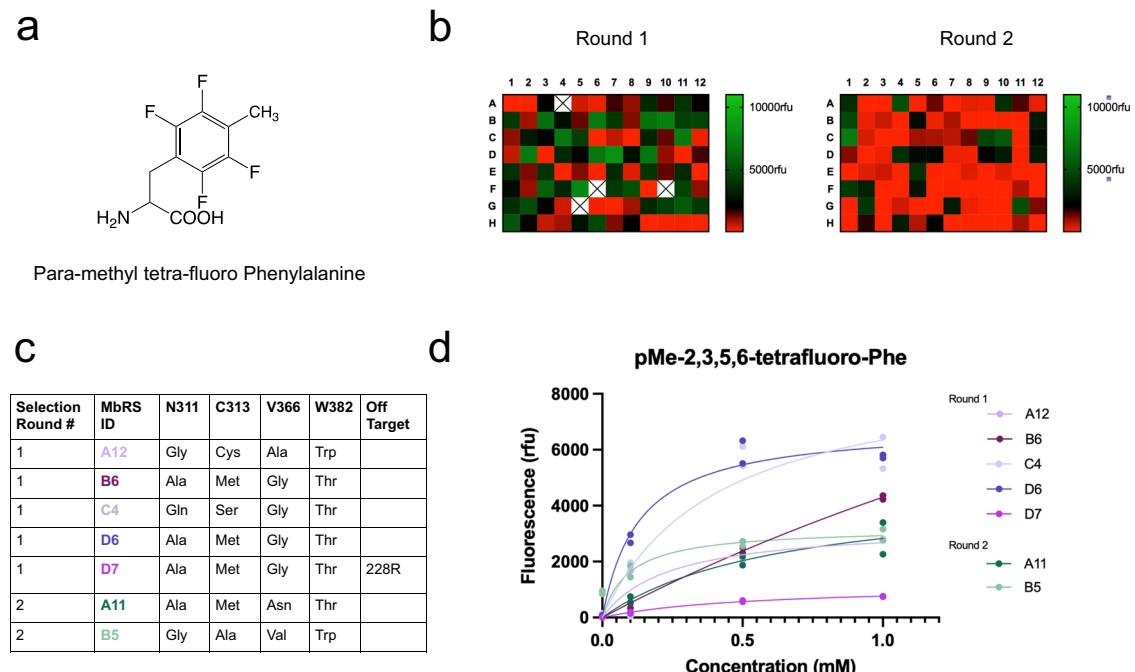

**Fig. 2 | Strategy and identification of synthetases. a** Chemical structure of para-methyl, tetra-fluoro Phe, which was used for screening in a PylRs library in *E. coli*. **b** Heatmaps of synthetases hits and relative GFP N150TAG rescue with para-methyl, tetra-fluoro Phe. Rounds 1 and 2 were independent screens of the library with para-methyl, tetra-fluoro Phe. **c** Table of the active site and off-target mutations borne by the top performing identified synthetases. MbRS indicates aminoacyl synthetases from the species *Methanosarcina barkeri*. **d** Potency (UP50) curves for the synthetases detailed in panel (**c**), *n* = 2 independent experiments for all conditions.

| Selection Round # | MbRS ID | N311 | C313 | V366 | W382 | Off Target |
|---|---|---|---|---|---|---|
| 1 | A12 | Gly | Cys | Ala | Trp | |
| 1 | B6 | Ala | Met | Gly | Thr | |
| 1 | C4 | Gln | Ser | Gly | Thr | |
| 1 | D6 | Ala | Met | Gly | Thr | |
| 1 | D7 | Ala | Met | Gly | Thr | 228R |
| 2 | A11 | Ala | Met | Asn | Thr | |
| 2 | B5 | Gly | Ala | Val | Trp | |

determine the concentration of ncAA required to express ncAA-sfGFP at half its maximum expression level, called the Unnatural Protein 50% (UP50)[40], were performed for the synthetases using para-methyl, tetra-fluoro Phe (Fig. 2D). The UP50 values for para-methyl tetra-fluoro Phe for the selected tRNA/RS pairs validate they will function efficiently at 0.5 mM ncAA[40].

## Synthetases for para-methyl, tetra-fluoro-Phe encode fluorinated Phe analogs

An important purpose of the screening efforts was to identify synthetases that enable the encoding of fluorinated (but non-methylated) phenylalanine analogs within proteins. Therefore we tested for permissivity[41] of the selected synthetases against a diverse collection

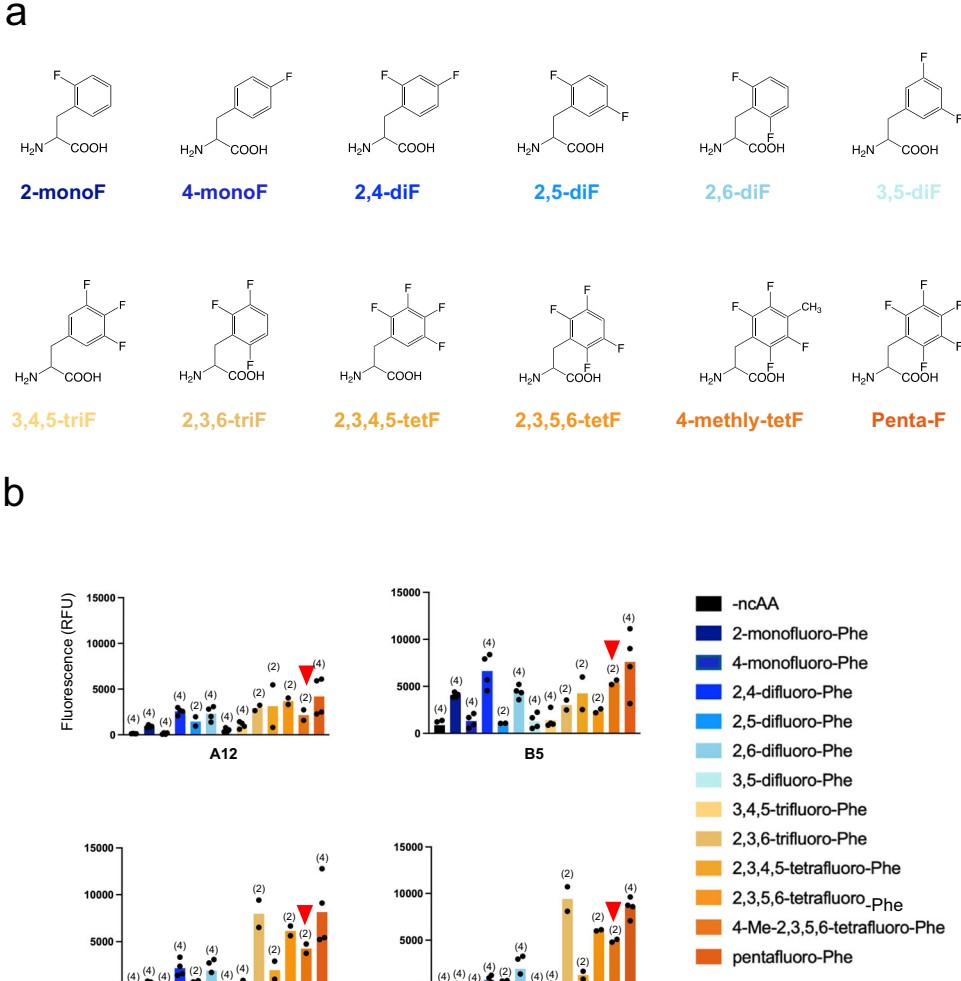

**Fig. 3 | Permissivity of the identified synthetases. a** Chemical structures of the fluorinated phenylalanine residues used in the screen. **b** Green fluorescence from the rescue of sfGFP-N150TAG in *E. coli* using 1 mM of each ncAA after 24-h incubation. N values of independent expressions in parentheses with bar height indicating the mean value. Red arrowheads indicate the level of GFP fluorescence for para-methyl, tetrafluoro-Phe, which was the Phe analog originally used for screening.

## Table 1 | Effects of fluorination on interaction energies with cations

| Phe_X substrate | Phe | 2F-Phe | 2,4F-Phe; 2,6F-Phe | 2,5F-Phe | 3,4,5F-Phe | 2,3,6F-Phe | 2,3,4,5F-Phe | 2,3,5,6F-Phe | 2,3,4,5,6F-Phe |
|---|---|---|---|---|---|---|---|---|---|
| Benzene analog | Benzene | 1F | 1,3F | 1,4F | 1,2,3F | 1,2,5F | 1,2,3,4F | 1,2,4,5F | 1,2,3,4,5F |
| | ΔG (kcal/mol) (% PHE) | ΔG (kcal/mol) (% PHE) | ΔG (kcal/mol) (% PHE) | ΔG (kcal/mol) (% PHE) | ΔG (kcal/mol) (% PHE) | ΔG (kcal/mol) (% PHE) | ΔG (kcal/mol) (% PHE) | ΔG (kcal/mol) (% PHE) | ΔG (kcal/mol) (% PHE) |
| Sodium | 26.7 (100) | 22.8 (85.4) | 19.1 (71.5) | 18.7 (70) | 16.2 (60.7) | 15.6 (58.4) | 12.7 (47.6) | 11.9 (44.6) | 9.1 (34.1) |
| Ammonium | 19.5 (100) | 16.3 (83.6) | 13.1 (67.2) | 13 (66.7) | 10.2 (52.3) | 10.1 (51.8) | 7.4 (37.9) | 6.8 (34.9) | 4.6 (23.6) |
| Guanidinium T-shaped | 14.8 (100) | 12.3 (83.1) | 10.3 (69.6) | 9.7 (65.5) | 8.1 (54.7) | 6.2* (41.9) | 3.9* (26.4) | 3.9* (26.4) | 1.8* (12.2) |

*Derived from the single-point calculation.

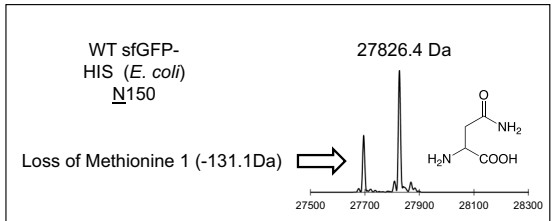

| ncAA | Δ WT mass (predicted) | Δ WT mass (observed) | |
|---|---|---|---|
| | | D6 | B5 |
| PENTA | 123 | 123.3 ± 1 Da | 123.3 ± 1 Da |
| 2,3,5,6F | 105 | 105.5 ± 1 Da | 105.4 ± 1 Da |
| 2,3,6F | 87 | 86.6 ± 1 Da | 87.1 ± 1 Da |
| 2,6F | 69 | 69.0 ± 1 Da | 69.4 ± 1 Da |
| 2F | 51 | N/A | 51.2 ± 1 Da |

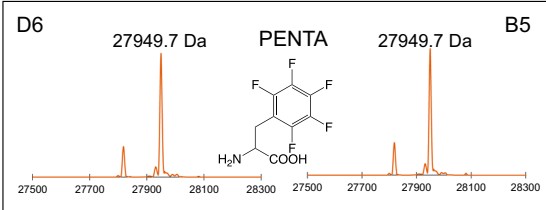

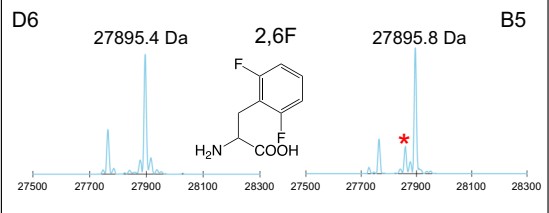

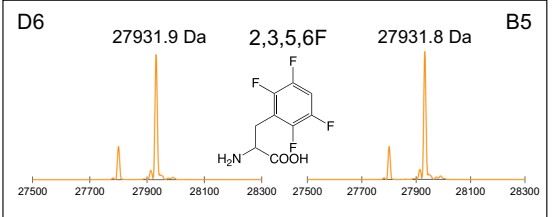

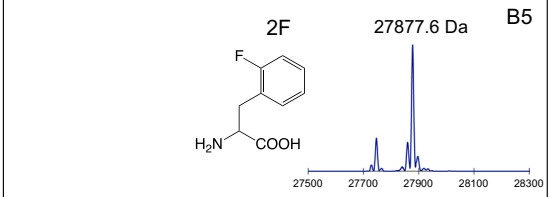

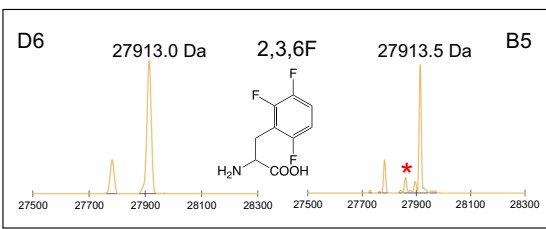

**Fig. 4 | ESI mass spectrometry confirmation of site-specific encoding of fluorinated Phe residues within superfolder GFP in *E. coli*.** sfGFP was purified via His tag from 50 mL growths. Table at top right: mass differences of sfGFP N150TAG from WT GFP conform to prediction for the molecular mass of each of phenylalanine analogs (Fig. 3a). fluoro Phe species. In some conditions for B5 synthetase, a small fraction of mass consistent with Phe incorporation was observed (asterisks). Mass spectrometry was not pursued for the combination of 2F Phe and D6 due to low expression in the screen.

In practice, this was an iterative process, with preliminary results from both *E.coli* (Fig. 3) and HEK cells (discussed below) motivating the synthesis and evaluation of additional structural analogs to audition in both systems. Interestingly, synthetases A12, C4, and D6 encoded penta-fluoro Phe and 2,3,5,6 tetra-fluoro Phe at comparable levels to para-methyl, tetra-fluoro Phe in *E. coli*, indicating that the para-methyl group is not required for recognition (Fig. 3b). Each synthetase showed some degree of permissivity; Phe$_{X-B5}$ and Phe$_{X-D6}$ were particularly interesting examples with substantially different recognition profiles (Fig. 3b). Phe$_{X-B5}$ displayed broader utility, but also lower apparent specificity over background than Phe$_{X-D6}$ in *E. coli*.

With permissivity profiles in hand, we performed ab initio quantum mechanical calculations to determine the effect of specific fluorination patterns on the potential of each analog to engage in electrostatic interactions, which has been previously done for a few of these analogs[25]. We calculated the theoretical ΔG (kcal/mol) for interactions with three different cations and their representative benzene structures (Table 1). As expected, increasing the number of fluorine atoms elicited a linear decrease in the binding energy. Table 1 expresses both the derived ΔG values and how they compare as a percentage of the native benzene of Phe (% Phe). Importantly, the relative positioning of the fluorine atoms did not greatly affect the energetics of binding, suggesting that the synthetases encode analogs with different potential local electrostatic effects but equivalent global perturbations. The encoding of multiple di-, tri-, or tetra-fluoro Phe analogs may prove useful to better understand phenylalanine side chains involved in complex, or multivalent motifs[25,42].

We next verified the fidelity of the top tRNA/RS pairs by analyzing the intact sfGFP proteins by ESI (electrospray ionization)–mass spectrometry to confirm the accurate encoding of the fluorinated phenylalanine derivatives[40]. We used the Phe$_{X-D6}$ and Phe$_{X-B5}$ synthetases to express and purify sfGFP_N150TAG_HIS in *E. coli*. Whole protein sfGFP masses were consistent with the replacement of N150 with each of the respective F-Phe ncAAs. Using Phe$_{X-D6}$, we observed evidence of high-fidelity incorporation (Fig. 4) for di-, tri-, tetra-, and pentafluoro phenylalanine (see Methods for additional details). When the Phe$_{X-B5}$ synthetase was used to express sfGFP bearing a range of ncAAs at N150, evidence of efficient encoding was clear for a wide range of fluorinated ncAAs (Fig. 4). However, for several ncAAs with lower encoding efficiency, detectable peaks corresponding to the mass of Phe at N150 were observed (Fig. 4, Δ mass of +33 Da from asparagine, asterisks). This background encoding of natural AAs was expected from the higher fluorescence values in the absence of ncAA for Phe$_{X-B5}$ (Fig. 3b), suggesting that Phe$_{X-B5}$ incorporates natural amino acids in *E. coli* when using ncAAs poorly recognized by the synthetase.

## Site-specific encoding of fluoro-Phe ncAAs into proteins in HEK cells

To evaluate the encoding of fluorinated analogs in HEK cells, we cloned the Phe$_{X-B5}$ and Phe$_{X-D6}$ synthetases into the pAcBac1[43] plasmid and co-transfected them with a plasmid encoding

## a

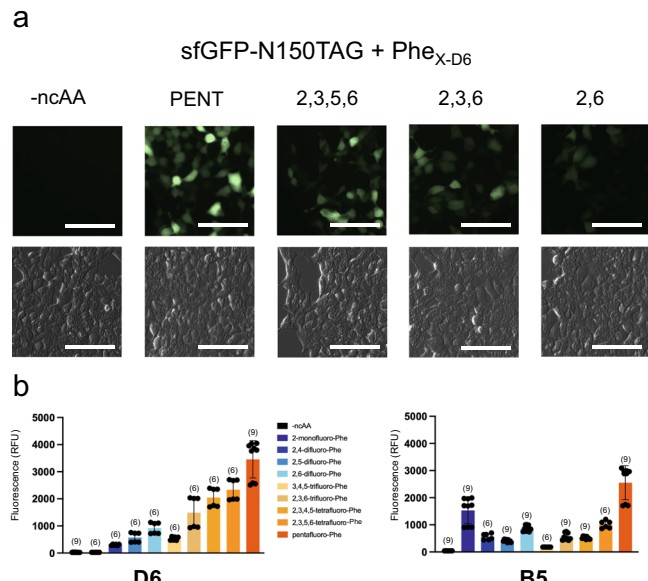

**Fig. 5 | Function of Phe_x synthetases in HEKT cells. a** Micrographs of HEKT cells co-transfected with GFP_N150TAG and Phe_X-D6 in the presence or absence of the given amino acid, GFP (top) and brightfield (bottom). In all cases, images were taken approximately 24 h after transfection, and the unnatural amino acid was used at a 2 mM concentration. Scale bar = 100 μm. Images are representative of four individual fields for no amino acid and penta fluoro-Phe, and two individual fields for each of the remaining amino acids. **b** Lysate GFP fluorescence for a range of fluorinated phenylalanine residues. *N* values are listed for a number of readings, with bar height indicating mean values ± standard deviation.

sfGFP_N150TAG in the presence or absence of fluoro-Phe ncAAs. In HEK cells, both the Phe_X-D6 and Phe_X-B5 tRNA/RS pairs produced sfGFP in the presence of fluoro-Phe ncAAs but not appreciably in their absence, yielding green cells which were imaged for fluorescence at 24 h (Fig. 5a). For quantitative comparison of ncAA-protein expression, fluorescence from cell lysate was measured. As with *E. coli* fluoro-Phe encoding, the position and number of fluorine substitutions affected the efficiency of producing ncAA-protein with the Phe_X-D6 and Phe_X-B5 tRNA/RS pairs (Fig. 5b). Of note, the relative fidelity of these tRNA/RS pairs was better in HEK cells compared to that in *E. coli*., but the background fluorescence from the Phe_X-B5 tRNA/RS pair was still higher than Phe_X-D6 (estimated 0.84% and 0.15% of maximum encoding for Penta-F Phe for Phe_X-B5 and Phe_X-D6 pairs, respectively).

To rigorously confirm the efficiency and fidelity of these tRNA/RS pairs in a eukaryotic context, ncAA-sfGFP was expressed in HEKT cells at 2 mM ncAA, purified via c-terminal His_6 tag and subjected to ESI−mass spectroscopy analysis. The Phe_X-D6 and Phe_X-B5 pairs encoded multiply fluorinated Phe analogs with high fidelity (Fig. 6). Estimated yield of sfGFP with Penta-fluoro Phe encoded by Phe_X-D6 was ~34 μg per gram of HEKT cells (Supplementary Fig. 1). From the *E. coli* permissivity evaluation we note a primary feature distinguishing Phe_X-B5 from Phe_X-D6 is the ability of the former to recognize 2-mono fluoro Phenylalanine. However, in HEKT cell expression of sfGFP-N150TAG with Phe_X-B5 and 2-mono fluoro Phe, we observed evidence of widespread encoding of 2-mono fluoro Phe at Phe-codons (Supplementary Fig. 2). This observation suggests that the endogenous (human) Phe tRNA/RS pair cannot effectively distinguish between 2-mono F Phe and Phe, preventing site-specific encoding of 2-monofluoro Phe in HEKT cells and likely other human cells as well. Overall, the ESI−mass spectroscopy analysis of the pure ncAA-protein demonstrates the fidelity of the Phe_X RS group to encode 6 of the fluorinated ncAAs; 2 di-fluorinated (2,5/2,6), a tri-fluorinated (2,3,6),

two tetra-fluorinated (2,3,5,6/2,3,4,5), and the penta-fluorinated phenylalanine in HEKT cells with high efficiency and fidelity. Dose-response experiments with ascending concentrations of the ncAAs indicate that a few combinations (such as Phe_X-D6 + penta-fluoro Phe) are useable at sub-millimolar concentrations of ncAA but that, in general, 1–2 mM is necessary for efficient ncAA-protein expression (Supplementary Fig. 3).

### Use of Phe_x synthetases to encode fluoro-Phe ncAAs into human proteins

Finally, we demonstrate the utility of the tRNA/Phe_x pairs for fluoro-Phe ncAA encoding within two large, human ion channels of high clinical significance. Mutations within the ~150 kDa cystic fibrosis (CF) transmembrane conductance regulator (hCFTR) cause the life-shortening disease CF (Fig. 7a)[44], while mutation and dysregulation of the ~200 kDa cardiac sodium channel (hNa_v 1.5) are associated with various cardiovascular diseases including cardiac arrythmia and heart failure (Fig. 7b)[45,46]. These proteins have been studied in a wide variety of eukaryotic cell hosts, but mammalian cell expression is either essential or preferred for many types of functional, biochemical, and structural analyses of channel regulation or pharmacology[47–60]. We introduced TAG codons at position F508 in hCFTR and at F1486 in hNa_v 1.5 and co-transfected these expression plasmids with Phe_X-D6 in the presence or absence of 2,3,6F-Phe. We harvested the cells two days post transfection and assessed expression via a western blot of cell lysates. In both cases, expression of full-length channels (as indicated by antibodies specific for C-terminal epitopes) was dependent on the addition of 2,3,6F-Phe in the growth media (Fig. 7a–c), demonstrating specific encoding by Phe_X-D6. Independent confirmation of 2,3,6F-Phe encoding within hNa_v 1.5 was accomplished via expression of hNa_v 1.5 F1486 (2,3,6F-Phe) via Phe_X-D6, purification via FLAG TAG and in-gel tryptic digestion/mass spectrometry (MS)− determination (Fig. 7d). Finally, WT and hNa_v 1.5 F1486 (2,3,6F Phe) channels were expressed in HEK cells and recorded via the whole-cell patch clamp configuration. In agreement with the western blot results, robust macroscopic currents were observed for both WT and mutant channels; these channels appeared to activate and inactivate normally in response to membrane depolarization (Fig. 7e). Analysis of gating parameters revealed that encoding 2,3,6F-Phe at position F1486 caused a subtle enhancement of channel inactivation. This was apparent in a small (~7 mV) but statistically significant left shift in the midpoint of steady-state inactivation (Fig. 7g, Supplementary Table 1). Interestingly, data from exhaustive natural mutation of this critical phenylalanine previously revealed a strong correlation between hydrophobicity and efficiency of inactivation; no natural mutation fully reproduced WT (Phe) behavior[61]. The data here suggest that the interaction of this side-chain within the inactivated conformation of channel[53] is enhanced via near-isosteric fluorine substitution on the benzene ring of F1486. This is consistent with studies of model hydrophobic cores wherein targeted fluorination enhanced thermodynamic stability over native levels[42].

## Discussion

The electrostatic contributions of aromatic residues to protein–protein interactions and ligand recognition are becoming increasingly appreciated[12], but available tools limit the ability to test proposed mechanistic hypotheses. The tRNA/RS pairs reported herein allow for the encoding of a wide spectrum of fluorinated phenylalanine residues, which can be used to tune the electrostatic contributions of the aromatic side chain in ππ- and cation-π interactions (Fig. 1, Table 1). The majority of the fluorinated phenylalanine residues used in this study are commercially available, making this tool set immediately adoptable for those studying aromatic side change contributions in prokaryotes and eukaryotes. As these tRNA/RS pairs were first evolved for a multiply fluorinated para-methyl-Phe analog, it is not surprising that the more highly fluorinated species tend to encode most efficiently into proteins

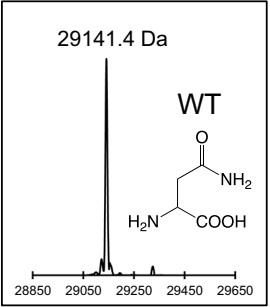

| ncAA | △ WT mass (predicted) | △ WT mass (observed) | |
| --- | --- | --- | --- |
| | | D6 | B5 |
| PENT | 123 Da | 122.6 ± 1 Da | 122.2 ± 1 Da |
| 2,3,5,6F | 105 Da | 104.7 ± 1 Da | 104.9 ± 1 Da |
| 2,3,4,5F | 105 Da | 104.7 ± 1 Da | 104.9 ± 1 Da |
| 2,3,6F | 87 da | 87.2 ± 1 Da | 86.9 ± 1 Da |
| 2,6F | 69 Da | 68.9 ± 1 Da | 68.9 ± 1 Da |
| 2,5F | 69 Da | 68.9 ± 1 Da | 68.8 ± 1 Da |

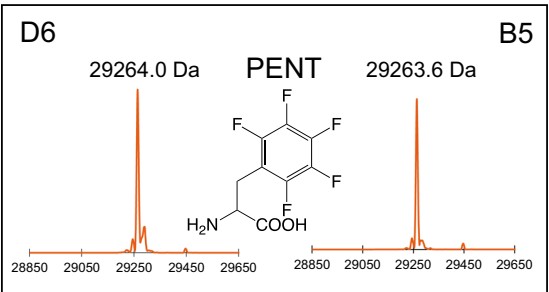
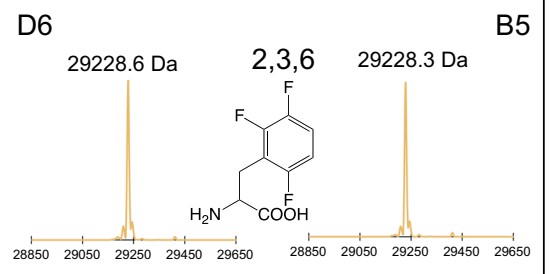

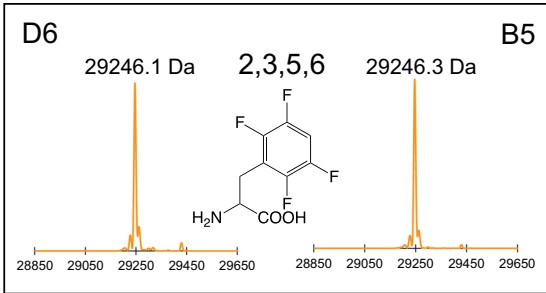
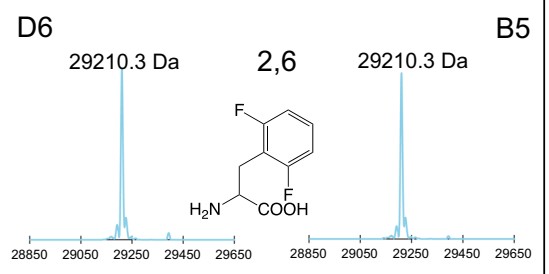

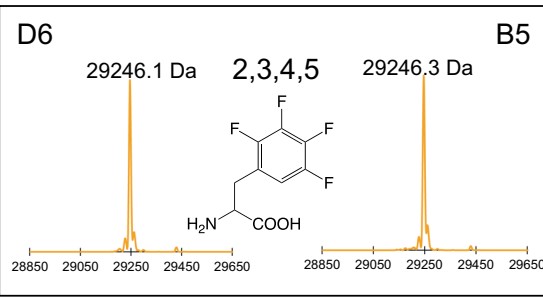
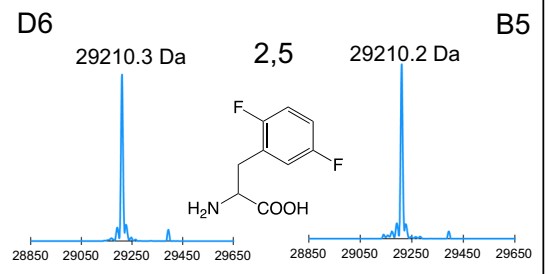

**Fig. 6 | ESI mass spec confirmation of encoding of fluorinated Phe analogs within sfGFP expressed in HEKT cells.** ncAAs were added at 2 mM concentration and cells were harvested approximately 36 h post transfection. sfGFP-WT-V5-HIS or N150_TAG-V5-HIS was purified via HIS tag. Table (at top right) reports mass differences of sfGFP N150TAG from WT GFP, which conform to the prediction for the molecular mass of each fluoro Phe species.

(Figs. 3 and 5). We view the fact that penta-fluoro and tetra-fluoro phenylalanine analogs encode so well as fortuitous because it enables the investigator to gain valuable early information on the electrostatic role of Phe residues given the strong electrostatic perturbation imparted by the fluorines. For example, penta-fluoro phenylalanine is estimated to bear only ~12–34% of the cation-π binding potential of native Phe (Table 1), whereas tri-fluorinated phenylalanine bears ~40–60%, depending on the cation involved.

The efficiency of the synthetases, as demonstrated herein, suggests that they will be useful for the site-specific encoding of non-canonical phenylalanine analogs within proteins in *E. coli* and HEK cells at levels sufficient for biochemical characterization. In HEK cells, the yield of the most robust combination (Phe$_{X-D6}$ and penta-fluoro phenylalanine) was 34 μg per gram of cell pellet using transient transfection. This is comfortably within the range necessary for structure determination by cryo-EM, provided the protein is efficiently purified and the growth is sufficiently scaled. It is worth noting that the PylRS system is useful in other mammalian cell lines beyond HEK[62]. That is, given the broad orthogonality of the *Mb* PylRS/Pyl-tRNA platform, we anticipate the F-Phe incorporation systems described here will be functional across other eukaryotic systems for which Genetic Code Expansion systems have been described. It is noted, however, that such applications will require additional levels of optimization relating to RS, tRNA, and target protein expression levels, as well as ncAA concentrations used in culture media.

The system may also be further optimized via the construction of synthetase-incorporated baculoviruses[43] or stable cell lines. The

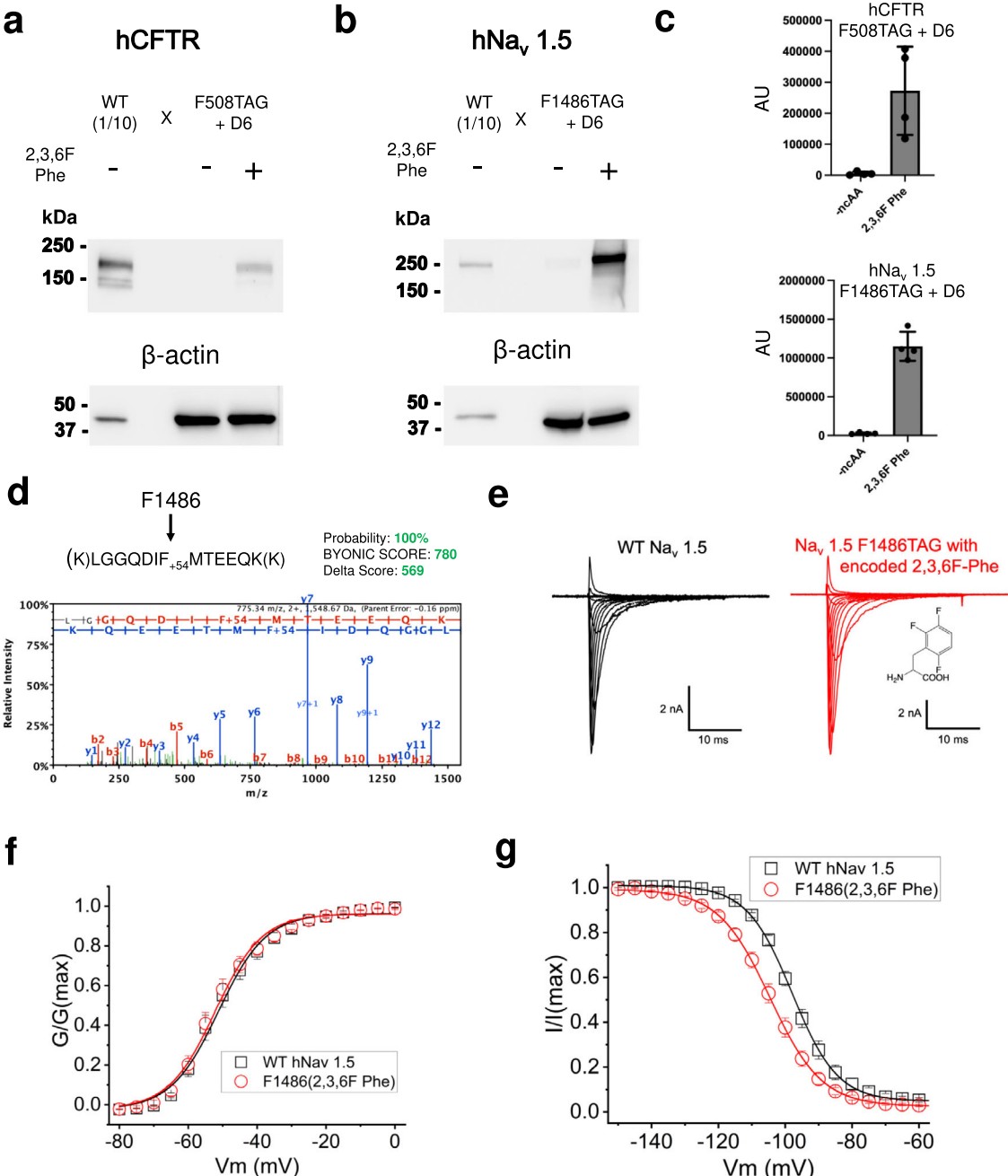

**Fig. 7 | Site-specific encoding of tri-fluoro-Phe within two large, human ion channels. a** Western blot of hCFTR using C terminal antibody (ab596) of WT CFTR or F508TAG-CFTR co-transfected with Phe$_{X-D6}$ in the presence or absence of 2 mM 2,3,6F Phe. WT lysates are loaded at 1:10 of TAG conditions. Uncropped blots are shown in the Supplementary Information file. **b** Western blot of hNa$_v$1.5 using C terminal antibody (D9J7S) of WT hNa$_v$1.5 or F1486TAG-hNa$_v$1.5 co-transfected with Phe$_{X-D6}$ in the presence or absence of 2 mM 2,3,6F Phe. Images are representative of four individual blots. Uncropped blots are shown in the Supplementary Information file. **c** Quantification (densitometry) of signal from 60 μg of the loaded lysate ($n = 4$ blots for each condition). **d** Mass spectra of a tryptic fragment from expressed and purified hNa$_v$1.5 F1486 (2,3,6F Phe) via D6. **e** Example current families for whole (HEK) cells expressing WT or mutant channels. Cells held at −140 mV and activated via depolarization in 5 mV increments. **f** Activation (G/V) curves for WT and mutant channels, $n = 5$ independent cells for wild-type and eight independent cells for hNa$_v$1.5 F1486 2,3,6F Phe. **g** Steady-state inactivation curves for WT and mutant channels, $n = 5$ independent cells for wild-type and eight independent cells for hNa$_v$1.5 F1486 2,3,6F Phe.

function of the synthetases in mammalian cells enables their users to answer questions that were previously intractable with other systems (prokaryotic, cell-free, *Xenopus* oocyte) and, in principle, can be extended to express and characterize myriad membrane and soluble proteins. Beyond mechanistic work, the synthetases may be used to site-specifically incorporate fluoro-phenylalanine nuclear magnetic resonance probes[63], and an optimized system utilizing these synthetases may enable the biotechnological production of therapeutic proteins wherein fluorinated phenylalanine residues are encoded for pharmacologically advantageous reasons[64].

## Methods

### Amino acids

All solvents and reagents were supplied by Sigma Aldrich and were used as-is unless explicitly stated. Dry nitrogen was supplied by Praxair and passed through two moisture-scrubbing columns

of dry calcium sulfate (Drierite) prior to use. High-performance liquid chromatography (HPLC) analyses were performed on a Waters 1525 Binary HPLC pump equipped with a Waters 2998 Photodiode Array Detector, employing Sunfire C18 analytical (3.5 μm, 4.6 mm × 150 mm, 0.8 ml/min) or preparative (5.0 μm, 19 mm × 150 mm, 10 ml/min) columns and Empower software. Buffers were drawn in linear gradients from 100% A (50 mM ammonium acetate) to 100% B (acetonitrile) over 30 min. Mass spectra were recorded on a Waters QToF Premier Quadrupole instrument, in both positive and negative modes. 2,3,5,6 tetra-fluoro Phe amino acid was synthesized according to a published procedure[42] with the following small modification: in lieu of ethanol extraction, following saponification of the methyl amide, the neutralized and lyophilized salt was purified directly via HPLC. The product was lyophilized to a white powder. (Calculated mass for $C_9H_7F_4NO_2$: 237.1 Da, found 238.0 Da (M + 1)/236.0 Da (M − 1) for MS in positive and negative modes). Synthesis of 2,3,5,6 tetrafluoro, paramethyl Phe was carried out according to the report of Mikaye-Stoner et al.[65] MS revealed a mass of 252.05 in positive mode and 250.06 in negative mode, which agreed with the predicted mass of 251.06.

The other Phe analogs were acquired commercially as follows: **2F** Phe -Astatech, cat # 73308; **4F** Phe- Chem-impex cat # 02572; **2,4F** Phe- 1ClickChemistry, cat #5C96757; **2,5F** Phe- Astatech, cat # 60350; **2,6F**- Phe HCl salt- Chem-impex, cat #24171; **3,5F** Phe Chem-impex, cat # 04123; **2,3,6F**-Phe- Astatech, A50355; **3,4,5F** Phe- Chem-impex, cat # 07394; **2,3,4,5F** Phe- Enamine, cat # en300-27751009; **2,3,4,5,6F** Phe- Chem-impex, cat # 07183.

### PylRS library screen in *E. coli*

A plasmid library composed of the 20 canonical amino acids ran-domized at 5 sites (Asn 311, Cys 313, Val 366, Trp 382, and Gly 386) was used alongside positive (pREP-pylT) and negative selection (pYOBB2-pylT) plasmids to select for aminoacyl-tRNA synthetases (RS) unique to pMe-2,3,5,6-tetrafluoro-Phe. The specific library used here has been used previously by our group[66]. This library was built off the wild-type *M. barkeri* pyrrolysine synthetase with five sites (N311, C313, V366, W382, and G386) randomized to all 20 amino acids using NNK degenerate codons. In most of our selections to date, including those described in this manuscript, G386 remains unchanged, indicating it that is not tolerant to alteration while still maintaining synthetase function. Indeed, for all our described hits here, residue 386 is conserved to Gly. Because of this invariance, we chose to leave it out of the Table in Fig. 2. All incubations for selec-tion are carried out at 37 °C for 16 h unless otherwise noted. All recovery steps last 1 h and are shaken at 250 rpm. Cultures are grown at 37 °C for 16 h and shaken at 250 rpm.

For positive selection, 500 μL of DH10B electrocompetent cells containing the positive pREP-pylT selection plasmid were transformed by electroporation with 1 μg of library plasmid. Cells were recovered in SOC media at 37 °C for 1 h. Recovered cells were serially plated from 10 to $10^{-6}$, and grown for 16 h at 37 °C to insure proper library coverage. Coverage was calculated to ~150×, representing >99% of possible library members. The pooled recovery was used to inoculate 500 mL of LB media with 50 μg/mL of Kanamycin (Kan) and 25 μg/mL of tet-racycline (Tet). The culture was grown overnight to saturation. This saturated culture was used to inoculate 500 mL of fresh LB media. This was grown to an $OD_{600}$ of 4.1, and 100 μL was plated on each of ten 15 cm LB-agar plates containing 50 μg/mL Kan, 25 μg/mL Tet, 40 μg/mL chloramphenicol (Cm), and 1 mM ncAA. Plates were incubated over-night. To harvest the resulting cells, 5 mL of LB media was added to each plate. Colonies were then scraped from plates, pooled, and recovered for 1 h. Cells were then pelleted, and plasmid DNA was extracted using a Macherey-Nagel miniprep kit. The resulting DNA was then plasmid separated by isolating the library plasmid on a 0.8%

agarose gel via agarose gel electrophoresis and extracting using the Thermo Scientific GeneJET gel extraction kit.

For negative selection, 100 ng of plasmid DNA from the positive selection was transformed into 50 μL of DH10B cells containing PYOBB2-pylT using electroporation and recovered in 1 mL SOC media. Following recovery, 100 μL was plated on each of 3 LB-agar plates containing 50 μg/mL Kan, 25 μg/mL Cm, and 0.2% arabinose. After 16 h, cells were scraped and DNA prepped as described above.

To identify functional synthetase variants, 100 ng of the pBK DNA from negative selection was transformed into 25 μL of electro-competent DH10B cells containing the reporter plasmid, pALS-pylT. This plasmid contains the sfGFP reporter with a TAG site at residue 150. In the presence of the selected ncAA, the TAG site will be suppressed, and the resulting colonies should appear green. Transformed cells were recovered in 1 mL SOC media. After recovery, cells were diluted by a factor of 100, and 100 μL of this dilution was plated on three 15 cm auto-inducing plates[40]. All plates contain 50 μg/mL Kan and 25 μg/mL Tet; two contain 1 mM ncAA and the other is left without ncAA as a control. Plates were incubated overnight. After 16 h, plates were kept at RT for 24 h to further develop. After fully maturated, colonies that appeared green were individually selected and used to inoculate 500 μL of non-inducing media[67] containing 50 μg/mL Kan and 25 μg/mL Tet in a 96-well block. This block was incubated for 20 h at 37 °C shaking at 300 rpm. After adequate growth, 20 μL of each well was used to inoculate two 96-well blocks with 500 μL of auto-inducing media[40,67] with 50 μg/mL Kan and 25 μg/mL Tet. One block contained 1 mM ncAA while the other did not. After 24 h of incubation (37 °C, 300 rpm), sfGFP fluorescence was measured. The 30 highest-performing synthetases were sequenced, and 17 unique sequences were identified.

### UP50 measurements

Electrocompetent DH10B cells containing the pALS reporter plasmid were transformed with isolated pBK plasmids containing the Phe$_X$ synthetases. The transformants of eight synthetases, A11, A12, B5, B6, C4, C10, D6, and D7, were used to inoculate five 500 μL cultures each, consisting of auto-induction media with 50 μg/mL Kan and 25 μg/mL Tet containing no ncAA, 0.1 mM, 0.2 mM, 0.5 mM, or 1 mM ncAA. Note that we subsequently realized that C10 and D6 had the same sequence and thus further characterized D6. These were performed in duplicate, and fluorescence and $OD_{600}$ were measured after 24 h.

### Permissivity screen (*E. coli*)

Electrocompetent DH10B cells containing the pALS reporter plasmid were transformed with isolated pBK plasmids containing the synthe-tases. Cultures of 5 mL were inoculated with single colonies of each synthetase in non-inducing media[40] with 50 μg/mL kanamycin and 25 μg/mL tetracycline and grown for 16 h. Noncanonical amino acid stocks were made at 100× their final concentration. Ninety-six-well blocks were prepared with 500 μL of auto-induction media containing antibiotics, and ncAAs with a final working concentration of 1 mM was added to their respective wells. Each well was then inoculated with a corresponding synthetase, yielding a final 96-well block where each synthetase was tested against each amino acid of interest. Cultures were grown in duplicate and fluorescence measurements were taken at 24 h. Previous work has shown that fluorination of aromatic residues encoded at position N150 does not substantially affect intrinsic sfGFP fluorescence[65].

### Expression and purification of sfGFP (*E. coli*)

Protein sequences of sfGFP variants used in this study are given in Supplementary Fig. 5. Previously existing *E. coli* containing sfGFP150TAG and a given RS in non-inducing media were used to inoculate 5 mL cultures of non-inducing media containing 50 μg/mL Kan and 25 μg/mL Tet. Cultures were incubated for 20 h at 37 °C,

shaking at 250 rpm. After adequate growth, 500 μL of the 5 mL cultures were used to inoculate 50 mL cultures of auto-inducing media containing 1 mM ncAA. Cultures were incubated for 24 h (37 °C, 250 rpm), after which cultures were then spun down at 5000 rcf for 10 min and resuspended in 10 mL of a Tris buffer solution containing 100 mM Tris, 0.5 M NaCl, and 5 mM imidazole. Cells were lysed by micro-fluidization at 18,000 psi and centrifuged at 20,0000 rcf for 30 min. sfGFP proteins in cell lysate were bound to TALON Metal Affinity Resin (bed volume of 100 μL) at 4 °C for 1 h. Lysate and bound resin were transferred to a gravity-flow column and the resin was washed with 30-bed volumes of buffer. Protein was eluted using the Tris buffer solution previously described supplemented with 200 mM imidazole. Protein concentration was assessed by comparing sfGFP fluorescence to a standard curve and submitted for ESI–MS.

As is apparent in the traces shown in Fig. 4, the deconvoluted mass spectra showed multiple masses for WT as well as mutant sfGFP. The spectra were analyzed in-house by Novatia (Promass) to yield relative intensities at given masses. In addition to the base (main) peak, both WT and mutant sfGFP displayed minor masses of ~−131 Da (lacking Methionine 1), ~−1378 Da, and ~−4165 Da. For the purpose of estimating encoding fidelity, these were considered part of the expected mass, and all other signals were considered non-expected mass. Note that the first time that sfGFP-N150TAG was expressed via $Phe_{X-D6}$ + 2,3,6F Phe, a minor +36.9 Da peak was present in the spectra. This was classified as a K+ adduct of the expected mass by Promass software. It was suggested to us that this mass may have, in fact, been due to contamination of this sample by sfGFP with penta-fluoro Phe encoded (as a result of technical error). We re-expressed the sfGFP-N150TAG via $Phe_{X-D6}$ + 2,3,6F Phe, and in this second purification, we did not observe the +36.9 Da peak. The trace for this sample is shown in Fig. 4. For $Phe_{X-D6}$, encoding fidelity for Penta-, 2,3,5,6-, 2,3,6-, and 2,6-fluoro Phe was 98.2%, 98.7%, 100.0%, and 95.0%, respectively. For $Phe_{X-B5}$, encoding fidelity for Penta-, 2,3,5,6-, 2,3,6-, 2,6-, and 2-fluoro Phe was 97.5%, 97.9%, 88.1%, 81.8%, and 95.6%, respectively.

## Mammalian cell culture
HEK 293T cells (CRL 3216) were maintained in DMEM high glucose medium supplemented with Pen/Strep, L glutamine, and 10% FBS (Sigma). Cells were used in passages 5–35.

## Construction of pAcBac1-Phe$_X$ Plasmid
The pAcBac1.tR4-MbPyl, used to express MbPylRS in mammalian cells, was a gift from Peter Schultz (Addgene plasmid # 50832). pAcBac1.tR4-MbPyl and relevant human codon optimized RS DNA fragments were digested with NheI/EcoRI restriction enzymes, ligated and transformed into NEB Stable cells. Plasmid integrity from individual clones was initially checked by analytical restriction digestion, and then confirmed by sequencing.

## Permissivity screen (mammalian)
HEKT cells were seeded in 60 mm dishes so that they would be approximately 60–80% confluent the next day. We transfected the cells using 1.75 μg of the PacBac1-based synthetase plasmid ($Phe_{X-D6}$ or $Phe_{X-B5}$) and 0.75 μg of sfGFP_N150TAG per dish. Polyjet was used (7.5 μl per dish, 2.5 mL media volume). Media were exchanged to contain ncAAs at a given concentration before transfection and approximately 16 h after transfection. Amino acids were solubilized either in NaOH or synthesized as free acids (penta-flouro, 2,3,6 trifluoro, 3,4,5 trifluoro, 2,5 difluoro, 2,4 difluoro, 2 monofluoro) or in HCl for those synthesized as HCl salts (2,3,5,6 tetrafluoro, 2,3,4,5 tetrafluoro, 2,6 difluoro). Cells were imaged and harvested approximately 24 h after transfection. Images were taken with 20× objective using a Leica DFC9000GT camera with 700 ms exposure, without binning. Excitation was through an X-Cite 120 LED (Lumen Dynamics) set to

100% intensity. Images were taken in monochromatic mode using a GFP filter (Chroma, Inc. filter #41017) and pseudocolored to green. For bright field images, the light was adjusted manually. To harvest, cells were washed twice with ice-cold DPBS, then sloughed off the dishes in 1 mL ice-cold DPBS supplemented with Roche protease inhibitors. Cells were collected in microcentrifuge tubes and pelleted by centrifugation. The supernatant was removed and the cell pellets were flash-frozen in liquid nitrogen, then stored at −80 °C. To lyse, 350 μl of RIPA buffer (Sigma) plus Roche protease inhibitor tablets were added to each pellet. After cell debris and nuclei were cleared via centrifugation, GFP fluorescence was read from supernatants in 96-well plates; for each concentration of amino acid, at least six readings were made from at least two transfections. The level of fluorescence from lysates from untransfected cells averaged 21.7 ± 2.1 RFU. By comparison, cells transfected with sfGFP_N150TAG and $Phe_{X-D6}$ averaged 26.9 ± 5.6 RFU, and $Phe_{X-B5}$ averaged 43.1 ± 2.5. The maximum background signal thus attributable to $Phe_{X-B5}$ and $Phe_{X-D6}$ was thus ~5 RFU and ~21.4 RFU, respectively.

## Expression for purification and MS (mammalian)
HEK 293 T cells were expanded in 10 mm dishes. For a given synthetase/amino acid combination, the size of the transfection (the number of plates) was estimated from differences in relative sfGFP_N150TAG yield in the multi-specificity screens, with poorer encoders requiring more cell mass. Overall, 7–40 dishes were transfected with master mixes such that each dish received 3.5 μg of pACBAC1-Phe$_x$, 1.5 μg sfGFP_N150TAG, and 15 μL of PolyJET according to the manufacturer's instructions. Media was changed before transfection and approximately 18 hrs after transfection. Cells were harvested approximately 36 h after transfection. To harvest, each dish was washed twice with DPBS. Cells were harvested via scraping in ice-cold DPBS plus Roche protease inhibitors, on ice, pelleted by centrifugation, and flash-frozen in liquid nitrogen. Cells were lysed in hypotonic lysis buffer (10 mM Tris/HCl pH 8, Roche protease inhibitors, 0.1% Triton X 100 Sigma) via dounce homogenization on ice. Cell debris and nuclei were cleared via centrifugation. Clarified lysate was diluted 1:2 in Wash buffer (25 mM Tris/HCl pH 8, 20 mM Imidazole, 150 mM NaCl) and applied to pre-equilibrated Ni-NTA resin (Qiagen). The column was washed in at least 30 volumes of wash buffer, then the protein was eluted in elution buffer (25 mM Tris/HCl pH 8, 250 mM Imidazole, 150 mM NaCl). Elution was exchanged into 100 mM TEAB, prepared via 1:10 dilution of 1 M Stock (Thermo Fischer) in ultrapure water. Finally, the protein was concentrated to ~0.5 mg/ml, flash frozen, and submitted to Novatia Inc for ESI mass spec of the intact protein. The LC-UV traces from Novatia indicated that purified WT and mutant sfGFP existed as three predominant isoforms, a dominant peak, and two minor peaks representing +532 ± 1 Da and −42 ± 1 Da mass differences (Supplementary Fig. 4). Note that the WT condition was co-transfected with $Phe_{X-B5}$ synthetase, which did not affect the mass (within <1 Da of the same WT sfGFP expressed alone[40]). As was also the case when expressed in *E.coli*, the deconvoluted mass spectra for the dominant LC peak showed multiple masses for WT, as well as mutant sfGFP, from HEK cells. The spectra were analyzed in-house by Novatia (Promass software) to yield relative intensities of the dominant mass and any other apparent peaks. WT and mutant sfGFP shared minor peaks of ~−42 Da (appx. 2% of integration), ~−1288 Da (appx. 2% of integration) -183 Da (appx. 3% of integration), and ~+53 Da (appx. 1% of integration) in common and thus, for the sake of estimation of encoding fidelity, these were counted among the expected mass. To be conservative, all other peaks were considered unexpected mass, even though they may not represent actual misincorporation. Fidelity for $Phe_{X-D6}$ and Penta-, 2,3,4,5-, 2,3,5,6-, 2,3,6-, 2,6-, and 2,5-fluoro phenylalanine were estimated to be 97.5%, 100%, 98.2%, 98.5%, 97.6%, and 95.6%, respectively. For $Phe_{X-B5}$, fidelity was 96.2%, 97.4%, 100%, 100%, 100%, and 91.3% for the same species.

To accurately estimate sfGFP yield in a scenario wherein loss during purification is minimized, we transfected and harvested 31-10 cm dishes of HEKT cells with Phe$_{X-D6}$ and sfGFP_N150TAG in the presence of 2 mM penta-fluoro Phenylalanine. sfGFP was purified as described above. Percent sfGFP was quantified using band detection and densitometry in Biorad Imagelab software (68% of lane signal).

## Expression of ion channel constructs with the site-specific encoding of 2,3,6F Phe

pCMV-CFTR was a gift from Paul McCray (University of Iowa). WT CFTR-TAA (the existing TAG stop codon changed to TAA), F508 TAG CFTR (TAG amber codon introduced at F508 in the above WT construct), as well as WT and F1486TAG hNav 1.5_strep_HIS_strep were produced using standard methods and sequenced through the open reading frame. For expression of WT CFTR, 10 cm dishes of HEK 293 T cells were transfected with 1.5 μg WT CFTR-TAA and 0.25 μg WT GFP. For expression of WT hNav 1.5, 10 cm dishes of HEK 293 T cells were transfected with 1.5 μg of WT hNav 1.5_strep_HIS_strep and 0.25 μg WT GFP. For encoding of 2,3,6 F Phe, 10 cm dishes of HEK 293 T cells were transfected from master mixes of DNA such that each dish received 3.5 μg of Phe$_{X-D6}$ and 0.25 μg of sfSFGFP_N150TAG. Depending on condition, 2 μg of F508TAG-CFTR or 1.5 μg of F1486TAG-Nav 1.5-SHS were added. Media was changed twice (-16 h and -32 h post transfection), and cells were harvested ~44 hrs post transfection and flash frozen in aliquots. Cell pellets were resuspended in RIPA (Sigma Aldrich) plus Roche protease inhibitors. Totally, 60 μg of protein from cleared, unconcentrated lysate (or a dilution as indicated) was loaded on 4–20% sodium dodecyl sulfate (SDS) gels. For Western blotting, gels were transferred to nitrocellulose membranes and probed with anti-CFTR (ab596 from Cystic Fibrosis Foundation, 1:1750) or anti-Nav 1.5 (D9J7S, Cell Signaling, 1:5000) overnight in 5% milk in TBST. Blots were washed, probed with HRP secondaries (1:10,000), and imaged using Clarity ECL (Biorad). Blots were subsequently probed for beta-actin using HRP-conjugated primary (AC-15, Novus Biologicals 1:2000) appx 3 h in 5% milk in TBST. Blots were washed and imaged using Clarity ECL (Biorad). Densitometry (background subtracted integrated density) was done in ImageJ.

For patch clamp recording of hNa$_v$ 1.5, in parallel, we co-transfected 10 cm dishes with 3.5 μg of Phe$_{X-D6}$, 0.25 μg of sfGFP_N150TAG, and 1.5 μg of either WT hNa$_v$ 1.5 strep_HIS_strep or hNa$_v$ 1.5 F1486TAG strep_HIS_strep. Both conditions were cultured in 2 mM 2,3,6 trifluoro Phe throughout. Media was changed at 16 hrs and again when seeding onto 35 mm dishes for recording. Recordings were done approximately ~42 h post transfection. The pipette solution contained 105 CsF, 35 NaCl, 10 EGTA, and 10 HEPES (pH 7.4). The bath contained (in mmol/L) 150 NaCl, 2 KCl, 1.5 CaCl$_2$, 1 MgCl2, and 10 Hepes (pH 7.4). Pipette resistances were around 2 mΩ and series resistance was compensated ≥85%. For a generation of $I/V$ curves, cells were held at −140 mV and pulsed in 5 mV depolarizing increments. For measurement of steady-state inactivation, cells were held at −140 mV and conditioned for 500 ms in 5 mV depolarizing increments. The test pulse was at −30 mV. Normalized G/V and SSI curves were fit to Boltzmann functions using Origin software.

For purification of hNa$_v$ 1.5 F1486(2,3,6 F Phe), ten 10 cm dishes of HEKT cells were co-transfected so that each dish received 1.5 μg pCDNA hNa$_v$ 1.5 F1486TAG_FLAG, 3.5 μg Phe$_{X-D6}$, and 0.25 μg sfGFPN150TAG plasmids. Media changes were as described for the western blots. Channels were solubilized and purified similar to the preparation used previously to obtain the cryo-EM structure of the cardiac channel[47], with the following modifications. First, we used DDM/CHS as a detergent throughout rather than exchanging for GDN during washes. Secondly, the elution was concentrated and run directly on SDS page gel instead of being subjected to size exclusion chromatography. Gel band excision, tryptic digestion, and MS/MS were performed by the Iowa Proteomics Core using standard methods.

## Quantum mechanical calculations of cation-pi binding potential

All the models were prepared using GaussView 6 and all quantum chemical calculations were performed using Gaussian 16[68]. Full geometry optimizations were carried out at the M06/6-31 G(d,p) level of theory. In some fluorinated aromatic cases where the optimization resulted in geometries that were not considered cation-$\pi$ interactions, the binding energy was obtained using the single-point method, as done previously[25]. In this approach, full geometry optimization of the cation and the aromatic system is performed without the fluorination at the position which caused incompatible geometries upon optimization. Then fluorine is appended to the aromatic ring with a bond distance calculated from the aromatic system with fluorination at the desired position, which is optimized in isolation. The energy of this system is then calculated and used to find the binding energy in the single-point calculations.

## Reporting summary

Further information on research design is available in the Nature Portfolio Reporting Summary linked to this article.

## Data availability

All data from this study are included in this published article, associated source data file, and supplementary information document. Source data are provided in this paper.

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

## Acknowledgements

The authors thank members of the Ahern and Mehl lab for helpful discussions. The work was supported by National Institutes of Health (NIH) grants R01-GM128420 and NINDS R24 NS104617 (to C.A.A.), R01-GM123455 and P41-GM104601 (to E.T.). R.B.C. and R.A.M. were supported in part by the GCE4All Biomedical Technology Development and Dissemination Center supported by the National Institute of General Medical Science grant RM1-GM144227.

## Author contributions

D.T.I., E.T., R.B.C., R.A.M., and C.A.A. designed experiments, G.D.G, D.T.I., C.J.C., M.L.H., S.M., F.F., A.R., and J.D.G., performed experiments and analyzed the data; D.T.I and C.A.A. prepared an initial draft and all authors edited the paper.

## Competing interests

C.A.A., D.T.I., R.B.C., and R.A.M. are co-founders of Halide Biologics. The remaining authors declare no competing interests.
