## [Peer Review File · Nature Communications]

Tuning phenylalanine fluorination to assess aromatic contributions to protein function and stability in cellsREVIEWER COMMENTS

Reviewer #1 (Remarks to the Author):

In this article, Galles, Infield and colleagues report the development of new tRNA/ RNA synthetases pairs based on the pyrrolysine system that allow specific incorporation of different fluorinated Phe into proteins. This tool is developed primarily to dissect the role of cation-pi interactions in proteins, although the authors discuss some examples of other uses.

The trick to find tRNA/RS pairs with specific fluorinated Phe ncAAs incorporation capabilities was to screen a library of Pyl synthetases using para-methyl tetra-fluoro-phenylalanine. The authors characterize the evolved pairs using sfGFP fluorescence as a reporter for successful ncAA incorporation. They then check whether the tRNA/RS pairs are permissive for non-methylated fluorinated Phe (the actual purpose being to develop a tool for modulating aromaticity with sequential/ incremental fluorination of Phe). And the good news is that indeed they are: several evolved synthetases can be used to incorporate di- tri- tetra and pentafluorinated Phe. The authors characterize the best pairs for unnatural protein expression in E.coli and in HEK cells. They evaluate expression by measuring sfGFP fluorescence in cells, and also perform mass spectrometry on the purified GFP proteins to demonstrate ncAA specificity against Phe. The mammalian cells appear very robust in that regard, while some non-specific Phe incorporation was observed for E.coli.

Finally, they demonstrate the applicability of this approach on two proteins with pharmacological relevance, the CFTR chloride channel and Nav1.5 channel wherein phenylalanine mutations have been linked to disease phenotype. They demonstrate successful incorporation of a tri-fluorinated ncAA by restoring expression of both CFTR and Nav1.5 with an amber stop codon at the Phe location. They further demonstrate the specificity of such incorporation with MS on purified protein Nav1.5. To correlate these experiments with functional data, the authors perform electrophysiology measurements on Nav1.5, and show that the channel is active but displays a subtle change in activity (enhancement of activation).

This is excellent work, supported by rigorous and well-designed experiments. The only aspect that might be missing at the moment is a proper demonstration of the application of the tool. The authors go as far as doing the incorporation of a tri-fluorinated Phe ncAA in the channel Nav1.5, measure its activity with patch-clamp recordings but do not compare it with other fluorinated Phe, to actually test the role of aromaticity. A complete experiment would involve comparing at least two fluorinated Phe with different aromatic contributions, and also with a "classical" Phe to "other natural amino acids" mutant (that data is already out there I guess).

I otherwise have only minor comments.

1. Inconsistent labeling of "pyrrolysine-based aminoacyl-tRNA synthetase/tRNA pairs" : tRNA/RS pairs, tRNA/aminoacyl synthetase pairs - tRNA / aminoacyl tRNA-synthetase. I suggest introducing the abbreviation tRNA/ RS pairs then sticking to it.

2. Typos: p.5 sythetase and p8 encage

3. Table/ fig 1C: what does MbRS means?

4. Could the authors clearly identify the para-methyl tetra-fluoro-phenylalanine in the graphs of the permittivity experiments – using a totally different color for example? Since this was the amino acid used for screening, it would make sense to highlight it.

5. For people not working with Nav1.5 channel, the following sentence is cryptic “ ... that the interaction of the IFM particle for its inactivated-state receptor is enhanced via near-isosteric fluorine .substitution on the benzene ring of F1486”. What is IFM particle?

Reviewer #2 (Remarks to the Author):

The details of the interactions carried by aromatic side chains are challenging to study with conventional mutagenesis tools since mutations to non-aromatic residues disturb the interactions, as well as the size and volume of the site. In the present study, Galles GD, Infield DT, et al. take advantage of the evolution of pyrrolysine-based aminoacyl-tRNA synthetase/tRNA pairs to enable site-specific incorporation of a variety of fluorinated phenylalanine (Phe) analogs into proteins (GFP, hCFTR, and rNav1.5), presenting us a flexible yet powerful tool to address the finesse of pi-pi /ion-pi interactions. Moreover, they use the ability of the tRNA/synthetase pairs to function in cross-kingdoms (E Coli and HEK mammalian cells) as a proof of concept to make this approach suitable to use at a bigger scale for structural and biophysical studies.

While the methods section is very detailed, making it very useful for those who desire to use it as a protocol, the results section about the Pyl-based screen could use some more details to help the reader understand better the system used:

Did the authors use a new library, or one used previously? Which was the starting synthetase?

Position G386 (original PylRS, I assume) is mentioned in the Methods section as one of the five positions randomized on the screen. However, the information about the mutations at this position is missing from the Table in Fig 2. Was this position ultimately considered as part of the screening?

The authors successfully incorporate tri-fluoro-Phe by the D6 synthetase variant in two membrane proteins, the human Cl⁻ channel CFTR and the rat Na⁺ channel Nav1.5, as an ideal demonstration of how helpful this method can be in studying membrane proteins; however, functional data is missing on hCFTR, and it might be the perfect addition to complete this section of the results. Did the authors try to register HEK cells expressing hCFTR?

Minor points:

C10 is mentioned in the Methods section along with the top performant synthetases from Fig 2. Is it also a top performant synthetase? Is there data available for this one?

The authors claim that the 2,3,5,6-tetra-fluo Phe is encoded at comparable levels to penta-fluo phe and para-methyl tetra-fluoro Phe in E Coli (pg7). However, according to the results in Fig 3B, this statement is consistent with A12, C4, and D6 but not B5 if I'm reading the graph correctly. 2,3,5,6-tetra-fluo-Phe seems to be encoded at lower levels than 2,3,4,5-tetrafluo-Phe, penta-fluo, and the para-methyl tetra-Phe. Please adjust the statement.

Typo: Phe missing in the legend of Fig 3B at the end of 2,3,5,6-tetrafluoro

Reviewer #3 (Remarks to the Author):

Dr. Ahern and others report the generation of a series of pyrrolysine-based aminoacyl-tRNA synthetase/tRNA pairs that can recognize para-methyl, tetra-fluoro-phenylalanine as substrates. These fluorinated phenylalanines can be successfully incorporated into proteins in E. coli and mammalian cells. Furthermore, the authors demonstrated that human CFTR and Nav1.5 with fluorinated phenylalanine can be expressed at levels adequate for biochemical studies. This is a solid study. It will be of high interest to the community if the authors can address these comments: (1) it is essential for authors to demonstrate the use of these analogs in biochemical studies. In the introduction, the authors suggest the importance of these analogs in exploring cation-Pi, Pi-Pi, and other interactions. The story will be complete if the authors can indeed use fluorinated phenylalanines to explore these interactions; and (2) the structures of fluorinated phenylalanines are very similar to phenylalanine. Authors should evaluate and discuss the incorporation of these phenylalanines by endogenous phenyl -tRNA synthetase/tRNA pairs.

I have some other comments:

1. Authors choose F508 of CFTR and F1486 of hNav 1.5 for the incorporation of fluorinated phenylalanines. The authors should explain why these two residues were chosen. Other residues with cation-Pi, Pi-Pi, and other interactions should be screened.

2. full name of Pyl should be given.

3. Page 7, that the para-methyl group is not required for recognition.

The authors mentioned that para-methyl group is not necessary for recognition. In this case, why not choose non-methyl substrate, 2,3,5,6-tetF, to do screening?

4. Compound Names in Figure 3A and B, Table 1 and Figure4.

The authors use 2,3,5,6-tetF in Figure 3A, but the same compound was called 2,3,5,6-tetrafluoro in Figure 3B. In addition, it was named with 2,3,5,6F in Table 4 and Figure 4. The names should be consistent.

5. Supplemental Figure 1B.

The authors should label the lengths of related bands of ladder for target protein.

6. Figure 3 and Figure 5.

How many independent biological replicates? Three?

7. Authors claimed that the synthetases tested encoded penta-fluoro Phe and

2,3,5,6 tetra-fluoro Phe at comparable levels to para-methyl tetra-fluoro Phe in E. coli. This is not consistent with Figure 3B.

Reviewer #4 (Remarks to the Author):

This manuscript reports the genetic encoding of fluorinated phenylalanines in both bacterial and mammalian cells. Although the genetic incorporation of fluorinated phenylalanine derivatives has been reported before, this current work identified more efficient PyIRS mutants. Overall, the work is solid, and can be published with some revisions.

1. Fluorinated phenylalanines are structurally very similar to phenylalanine. It is interesting that the identified mutants are fairly specific. The authors may consider conducting docking studies, analyzing their data more deeply, and adding relevant discussions.
2. In Figure 3B, a positive control may be included, such as fluorescence intensity from wt sfGFP or from a well-established PyIRS mutant.
3. The PyIRS system has been applied to mammalian cells other than HEK. In the discussion section, the authors may provide a perspective on using the current system in other mammalian cells.
4. Add scale bars to Figure 5A.
5. Are these amino acids optically pure or racemic mixtures? Add stereochemistry to structures.
6. Some error bars are fairly large. Statistical significance may be determined and added.
7. If possible, it would be beneficial for others to use the system described in this work by providing plasmid maps in Supplementary Materials.

We thank the reviewers for their constructive criticisms of our manuscript. Our point-by-point response is below. We appreciate that the reviewers agree that the new synthetases that we have described and validated here will be of significant value to the research community. Reviewer's comments are in italics. Our responses are in blue font.

Reviewer #1 (Remarks to the Author):

In this article, Galles, Infield and colleagues report the development of new tRNA/ RNA synthetases pairs based on the pyrrolysine system that allow specific incorporation of different fluorinated Phe into proteins. This tool is developed primarily to dissect the role of cation-pi interactions in proteins, although the authors discuss some examples of other uses.

The trick to find tRNA/RS pairs with specific fluorinated Phe ncAAs incorporation capabilities was to screen a library of Pyl synthetases using para-methyl tetra-fluoro-phenylalanine. The authors characterize the evolved pairs using sfGFP fluorescence as a reporter for successful ncAA incorporation. They then check whether the tRNA/RS pairs are permissive for non-methylated fluorinated Phe (the actual purpose being to develop a tool for modulating aromaticity with sequential/ incremental fluorination of Phe). And the good news is that indeed they are: several evolved synthetases can be used to incorporate di- tri- tetra and pentafluorinated Phe. The authors characterize the best pairs for unnatural protein expression in in E.coli and in HEK cells. They evaluate expression by measuring sfGFP fluorescence in cells, and also perform mass spectrometry on the purified GFP proteins to demonstrate ncAA specificity against Phe. The mammalian cells appear very robust in that regard, while some non-specific Phe incorporation was observed for E.coli.

Finally, they demonstrate the applicability of this approach on two proteins with pharmacological relevance, the CFTR chloride channel and Nav1.5 channel wherein phenylalanine mutations have been linked to disease phenotype. They demonstrate successful incorporation of a tri-fluorinated ncAA by restoring expression of both CFTR and Nav1.5 with an amber stop codon at the Phe location. They further demonstrate the specificity of such incorporation with MS on purified protein Nav1.5. To correlate these experiments with functional data, the authors perform electrophysiology measurements on Nav1.5, and show that the channel is active but displays a subtle change in activity (enhancement of activation).

1. This is excellent work, supported by rigorous and well-designed experiments. The only aspect that might be missing at the moment is a proper demonstration of the application of the tool. The authors go as far as doing the incorporation of a tri-fluorinated Phe ncAA in the channel Nav1.5, measure its activity with patch-clamp recordings but do not compare it with other fluorinated Phe, to actually test the role of aromaticity. A complete experiment would involve comparing at least two fluorinated Phe with different aromatic contributions, and also with a "classical" Phe to "other natural amino acids" mutant (that data is already out there I guess).

We thank the reviewer for bringing up this important point. We value this suggestion. The goals of the current paper have been to 1) rigorously validate the encoding fidelity of these new synthetases and 2) demonstrate (with biochemistry and function) their ability to encode F-Phe derivatives into complex human proteins at scale. We agree with the reviewer that their application to a specific interaction study is an important next step in their use, and numerous such studies are currently ongoing. Indeed, requests have been made for these reagents based on our *BioRxiv* preprint of this manuscript, and we plan to disseminate these tools upon acceptance of our manuscript. Further, the work has been performed at part of a collaboration with the newly launched Genetic Code Expansion For All Center at Oregon State University. The sole purpose of this new Center is to efficiently disseminate reagents and training for the new tools described in our paper, as well as others that will follow. Thus, we anticipate many such functional studies to be produced (and re-produced) from other labs in the near future. Further, we note that the specific use of F-Phe derivatives in the study of aromatic interactions is a mature area of research and the application aromatic fluorination has been validated by multiple groups using functional and computational studies over the past 25 years. In this light,

the specific contributions of the current work are to democratize access to these tools for work as at scale in bacteria expression systems and a variety of human cells (see additional response below to R3).

I otherwise have only minor comments.

2. Inconsistent labeling of “pyrrolysine-based aminoacyl-tRNA synthetase/tRNA pairs” : tRNA/RS pairs, tRNA/aminoacyl synthetase pairs - tRNA / aminoacyl tRNA-synthetase. I suggest introducing the abbreviation tRNA/RS pairs then sticking to it.

We have implemented this comment into the manuscript.

3. Typos: p.5 sythetase and p8 encage

Corrected.

4. Table/ fig 1C: what does MbRS means?

MbRS refers to the synthetase (RS) family used, Methanosarcina barkeri. We have now defined this abbreviation Fig 2C in the legend.

5. Could the authors clearly identify the para-methyl tetra-fluoro-phenylalanine in the graphs of the permissivity experiments – using a totally different color for example? Since this was the amino acid used for screening, it would make sense to highlight it.

Thank you for this suggestion we have added a marking for this in the resubmitted manuscript.

6. For people not working with Nav1.5 channel, the following sentence is cryptic “... that the interaction of the IFM particle for its inactivated-state receptor is enhanced via near-isosteric fluorine substitution on the benzene ring of F1486”. What is IFM particle?

We appreciate this comment as our goal is to provide a manuscript that is inclusive of scientists from all backgrounds. We have clarified this passage with the following clarified sentence:

“The data here suggest that the interaction of this side-chain within the inactivated conformation of the channel [53] is enhanced via near-isosteric fluorine substitution on the benzene ring of F1486. This is consistent with studies of model hydrophobic cores wherein targeted fluorination enhanced thermodynamic stability over native levels [42].”

Reviewer #2 (Remarks to the Author):

The details of the interactions carried by aromatic side chains are challenging to study with conventional mutagenesis tools since mutations to non-aromatic residues disturb the interactions, as well as the size and volume of the site. In the present study, Galles GD, Infield DT, et al. take advantage of the evolution of pyrrolysine-based aminoacyl-tRNA synthetase/tRNA pairs to enable site-specific incorporation of a variety of fluorinated phenylalanine (Phe) analogs into proteins (GFP, hCFTR, and rNav1.5), presenting us a flexible yet powerful tool to address the finesse of pi-pi /ion-pi interactions. Moreover, they use the ability of the tRNA/synthetase pairs to function in cross-kingdoms (E Coli and HEK mammalian cells) as a proof of concept to make this approach suitable to use at a bigger scale for structural and biophysical studies.

We thank the reviewer for their positive comments and for appreciating the “flexible and powerful” nature of this new tool.

1. While the methods section is very detailed, making it very useful for those who desire to use it as a protocol, the results section about the Pyl-based screen could use some more details to help the reader understand better the system used:

Did the authors use a new library, or one used previously? Which was the starting synthetase? Position G386 (original PylRS, I assume) is mentioned in the Methods section as one of the five positions randomized on the screen. However, the information about the mutations at this position is missing from the Table in Fig 2. Was this position ultimately considered as part of the screening?

We thank the reviewer for identifying this discrepancy. We have now added the following descriptions of the library, synthetases, and strategy that was used in the Methods section, under PylRS library screen in E. coli: the library used here was one used previously (e.g. Jang et al. *J. Am. Chem. Soc.*, 2020, 142, 16, 7245–7249). This library was built off the wild-type *M. barkeri* pyrrolysine synthetase with five sites (N311, C313, V366, W382 and G386) randomized to all 20 amino acids using NNK degenerate codons. In most of our selections to date, including those described in this manuscript, G386 remains unchanged indicating it that is not tolerant to alteration while still maintaining synthetase function. Indeed, for all our described hits here, residue 386 is conserved to Gly. Because of this invariancy we chose to leave it out of the Table in Fig. 2, but in hindsight we can see the value of including it and so we have done so in these revisions.

2) The authors successfully incorporate tri-fluoro-Phe by the D6 synthetase variant in two membrane proteins, the human Cl- channel CFTR and the rat Na+ channel Nav1.5, as an ideal demonstration of how helpful this method can be in studying membrane proteins; however, functional data is missing on hCFTR, and it might be the perfect addition to complete this section of the results. Did the authors try to register HEK cells expressing hCFTR?

This is a valid point and agree that is worthy of future study. Encouragingly, the western blots in Figure 7 suggest that F508 2,3,6F Phe channels trafficked to the plasma membrane. This biochemical feature is apparent when comparing the banding pattern between this mutant and the WT CFTR channel.

Minor points:

C10 is mentioned in the Methods section along with the top performant synthetases from Fig 2. Is it also a top performant synthetase? Is there data available for this one?

Upon sequencing we discovered that C10 contained the same active site mutations as D6, which has been fully characterized in our study. This is now noted in the Methods section.

The authors claim that the 2,3,5,6-tetra-fluo Phe is encoded at comparable levels to penta-fluo phe and para-methyl tetra-fluoro Phe in E Coli (pg7). However, according to the results in Fig 3B, this statement is consistent with A12, C4, and D6 but not B5 if I'm reading the graph correctly. 2,3,5,6-tetra-fluo-Phe seems to be encoded at lower levels than 2,3,4,5-tetrafluoro-Phe, penta-fluo, and the para-methyl tetra-Phe. Please adjust the statement.

Thank you for pointing out this detail. We have refined the wording of this section to reflect the biological complications that are apparent within site-to-site comparisons for encoding efficiencies.

Typo: Phe missing in the legend of Fig 3B at the end of 2,3,5,6-tetrafluoro

Thank you, this has been corrected.

Reviewer #3 (Remarks to the Author):

Dr. Ahern and others report the generation of a series of pyrrolysine-based aminoacyl-tRNA synthetase/tRNA pairs that can recognize para-methyl, tetra-fluoro-phenylalanine as substrates. These fluorinated phenylalanines can be successfully incorporated into proteins in E. coli and mammalian cells. Furthermore, the authors demonstrated that human CFTR and Nav1.5 with fluorinated phenylalanine can be expressed at levels adequate for biochemical studies. This is a solid study. It will be of high interest to the community if the authors can address these comments:

(1) it is essential for authors to demonstrate the use of these analogs in biochemical studies. In the introduction, the authors suggest the importance of these analogs in exploring cation-Pi, Pi-Pi, and other interactions. The story will be complete if the authors can indeed use fluorinated phenylalanines to explore these interactions; and

See our reply to a similar concern by reviewer one. In particular.....

(2) the structures of fluorinated phenylalanines are very similar to phenylalanine. Authors should evaluate and discuss the incorporation of these phenylalanines by endogenous phenyl -tRNA synthetase/tRNA pairs.

This specific point is discussed in the relevant section which contains intact ESI mass spec data of GFP bearing F-Phe analogs. Indeed, in only one case we observed mass deviations consistent with the recognition of 2-mono-F Phe by the endogenous HEK Phe synthetase and subsequent encoding at Phe codons (Supp Figure 2).

I have some other comments:

1. Authors choose F508 of CFTR and F1486 of hNav 1.5 for the incorporation of fluorinated phenylalanines. The authors should explain why these two residues were chosen. Other residues with cation-Pi, Pi-Pi, and other interactions should be screened.

As discussed in the relevant section.....

2. full name of Pyl should be given.

We now introduce pyrrolysine and the Pyl abbreviation in the first sentence of the Results section.

3. Page 7, that the para-methyl group is not required for recognition. The authors mentioned that para-methyl group is not necessary for recognition. In this case, why not choose non-methyl substrate, 2,3,5,6-tetF, to do screening?

The point of screening with para-methyl was to select against synthetases with intrinsic Phe recognition, that is, the added methyl group was a negative discriminant.

4. Compound Names in Figure 3A and B, Table 1 and Figure 4. The authors use 2,3,5,6-tetF in Figure 3A, but the same compound was called 2,3,5,6-tetrafluoro in Figure 3B. In addition, it was named with 2,3,5,6F in Table 4 and Figure 4. The names should be consistent.

Thank you for this comment. We have now added "Phe" to the 2,3,5,6 tetrafluoro for Figure 3B. Shortening this nomenclature was required in Figure 3A, Table 4, and Figure 4 due to space requirements.

5. Supplemental Figure 1B. The authors should label the lengths of related bands of ladder for target protein.

Thank you, we have revised this figure to include ladder sizing.

6. Figure 3 and Figure 5. How many independent biological replicates? Three?

We now have added N-value details to both figure legends.

7. Authors claimed that the synthetases tested encoded penta-fluoro Phe and

2,3,5,6 tetra-fluoro Phe at comparable levels to para-methyl tetra-fluoro Phe in *E. coli*. This is not consistent with Figure 3B.

We now See R2 comment (second minor point)

Reviewer #4 (Remarks to the Author):

This manuscript reports the genetic encoding of fluorinated phenylalanines in both bacterial and mammalian cells. Although the genetic incorporation of fluorinated phenylalanine derivatives has been reported before, this current work identified more efficient PylRS mutants. Overall, the work is solid, and can be published with some revisions.

1. Fluorinated phenylalanines are structurally very similar to phenylalanine. It is interesting that the identified mutants are fairly specific. The authors may consider conducting docking studies, analyzing their data more deeply, and adding relevant discussions.

We thank the reviewer for this suggestion and agree with the value of understanding the basis for F-Phe selection over phenylalanine. Despite this motivation, detailed studies of synthetase-ncAA-tRNA remain quite challenging. None the less, we are attempting these studies part of a separate study on ncAA recognition mechanisms of orthogonal synthetases.

Technical challenges aside, we agree that it is tempting to speculate how the PylRS variants described here are able to select for F-Phe ncAAs over natural amino acids like Phe given their close structural similarity. We can only presume the relative positionings of electronegative fluorine atoms within the aromatic ring impart sufficiently unique electrostatic changes to be chemically distinguishable by the PylRS active site scaffold (e.g. Fig. 1B). But how these distinguishing features of F-Phe ncAAs are recognized by the PylRS variants will likely require structurally guided, quantum mechanically-based investigations, which could prove generally useful in the evolution of future substrate/enzyme interactions.

2. In Figure 3B, a positive control may be included, such as fluorescence intensity from wt sfGFP or from a well-established PylRS mutant.

We appreciate the nature of this inquiry but in our experience, the absolute encoding efficiency of a particular variant compared to GFP is not a useful predictor. Indeed, in Figure 7 we show that encoding within human ion channels produced yields on par with their WT counterparts. Thus, these new synthetases are amongst this most efficient that have been discovered for ncAA encoding.

3. The PylRS system has been applied to mammalian cells other than HEK. In the discussion section, the authors may provide a perspective on using the current system in other mammalian cells.

The following sentence has been added to the Discussion section: Given the broad orthogonality of the *Mb* PylRS/Pyl-tRNA platform, we anticipate the F-Phe incorporation systems described here will be functional in across other eukaryotic systems for which GCE systems have been described. It is noted, however, that such applications will require additional levels of optimization relating to RS, tRNA and target protein expression levels as well as ncAA concentrations used in culture media.

4. Add scale bars to Figure 5A.

Done.

5. Are these amino acids optically pure or racemic mixtures? Add stereochemistry to structures.

All of the amino acids used for genetic encoding are optically pure, L isoform, with one exception: the 4-methyl-terafuoro which is DL racemic.

6. Some error bars are fairly lager. Statistical significance may be determined and added.

Pairwise comparisons against are obviously highly statistically significant. We are not making claims within synthetases which show rescue over background as such comparisons may not be predictive of every encoding site and target protein of interest.

7. If possible, it would be beneficial for others to use the system described in this work by providing plasmid maps in Supplementary Materials.

Plasmid maps and construct sequences have been included in our resubmitted manuscript.

REVIEWERS' COMMENTS

Reviewer #1 (Remarks to the Author):

The authors have addressed all my concerns. I have no further comment, and I'm looking forward to seeing this published.

Reviewer #2 (Remarks to the Author):

I would like to thank the authors for addressing all my comments.

I just observed a typo on Figure 5B: Phe missing in the legend of the graph at the end of 2,3,5,6-tetrafluoro.

Other than that detail I have no further observations, and I look forward to seen published such an excellent work.

Reviewer #3 (Remarks to the Author):

The authors offered a detailed response to reviewers' comments and improved the manuscript. However, Reviewer 1 and 3's key suggestion that "a proper demonstration of the application of the tool is missing" has not yet been addressed. Without a proper demonstration of these fluorinated ncAAs in a biological model, it is hard to support the major claim that these analogs can be important to explore the residue interaction. If the goals of this paper are only to incorporate these analogs into complex human proteins, I would expect to be published in a journal specializing in protein engineering or Genetic Code expansion.

Reviewer #4 (Remarks to the Author):

After reading their revised manuscript, this reviewer felt that the comments were sufficiently addressed. The revised version is suitable for publication.

manuscript NCOMMS-22-28205

We thank reviewers for their time and expertise in reviewing our resubmitted manuscript. Our point-by-point verbatim response is below.

Reviewer's comments are in italics. Our responses are in blue font.

REVIEWERS' COMMENTS

Reviewer #1 (Remarks to the Author):

The authors have addressed all my concerns. I have no further comment, and I'm looking forward to seeing this published.

We thank the reviewer for their supportive comments.

Reviewer #2 (Remarks to the Author):

I would like to thank the authors for addressing all my comments.

I just observed a typo on Figure 5B: Phe missing in the legend of the graph at the end of 2,3,5,6-tetrafluoro.

Other than that detail I have no further observations, and I look forward to seen published such an excellent work.

We thank the reviewer for finding this typo and it has been corrected in our resubmitted manuscript.

Reviewer #3 (Remarks to the Author):

The authors offered a detailed response to reviewers' comments and improved the manuscript. However, Reviewer 1 and 3's key suggestion that "a proper demonstration of the application of the tool is missing" has not yet been addressed. Without a proper demonstration of these fluorinated ncAAs in a biological model, it is hard to support the major claim that these analogs can be important to explore the residue interaction. If the goals of this paper are only to incorporate these analogs into complex human proteins, I would expect to be published in a journal specializing in protein engineering or Genetic Code expansion.

While we understand this sentiment, the use of fluoro-aromatics to study protein function and stability is an established field, as are the accompanying computational methods. The major gap now filled by our study, is a means to efficiently perform these studies in vastly more types of proteins and expression systems.

Reviewer #4 (Remarks to the Author):

After reading their revised manuscript, this reviewer felt that the comments were sufficiently addressed. The revised version is suitable for publication.

We thank the reviewer for rereading the manuscript and for their supportive comments.